# Critical Thinking and Teacher Training in Secondary Education

**DOI:** 10.3390/jintelligence13030037

**Published:** 2025-03-13

**Authors:** Yasaldez Eder Loaiza, John Rodolfo Zona, Maria Fulvia Rios

**Affiliations:** 1Faculty of Arts and Humanities, Department of Educational Studies, University of Caldas, Manizales 170004, Colombia; 2Maria Goretti Educational Institution, Secretary of Education of Manizales, Manizales 170001, Colombia; rodolfo.zona@utp.edu.co; 3Sports Section, University Tecnology of Pereira, Pereira 660003, Colombia; fulvia.rioscpe2@gmail.com

**Keywords:** critical thinking, teacher training, conceptions, secondary education, argumentative processes

## Abstract

Various studies on the formation of critical thinking in teachers express difficulties in the training of teachers at different levels of education. Some of them recognize conceptual dispersion evidenced in their conceptions and explanations; others recognize curricular gaps without clear and coherent programs for their development; and a third group recognize that training in critical thinking requires both academic training as well as personal training of the subject. Most of the studies agree on training teachers from all fields of knowledge, where theoretical and methodological elements are provided to form critical thinkers. For this reason, the present research, developed with five teachers from different areas of secondary education in which different sources of information were collected and analyzed, contributes to the reflection with different theoretical perspectives and methodological strategies used by teachers, which were contrasted with the theories of critical thinking (psychological, philosophical and didactics of sciences).

## 1. Introduction

Theoretical developments on critical thinking have been studied from different disciplines, and each of them provides theoretical and methodological elements which consolidate their field of research, which currently leads to identifying, in the first instance, conceptual dispersion in that field of interest for current education ([35]; [27]; [37]; [34]; [39]). Second, the lack of knowledge, understanding and difficulties in their development by teachers of different educational levels ([23]; [37]; [34]; [20]) is an aspect of significant attention, taking into account that this type of thinking should become a relevant competence in teacher training ([24]).

Now, in the case of Colombia, critical thinking is adopted in formal education as a reference for quality; however, it lacks recognition in its epistemological, pedagogical and methodological foundations by limiting the teaching practice to the classroom, since it is affected by notions, beliefs and subjective perspectives of what it refers to. This issue led us to think about the need to work on the development of critical thinking with teachers, as a commitment to contribute to quality education from the classroom.

Regarding conceptual dispersion, contributions from different disciplines that converge in the field of critical thinking are analyzed ([34]; [37]). One of them, psychology, maintains that in the development of critical thinking, people make responsible decisions with the environment when different skills are deployed (interpretation, analysis, inference, explanation, evaluation and self-regulation), which according to [10] ([10]), are transferable to all fields of knowledge. In addition to this, the psychological perspective mentions as a second component a category called dispositions, related to the attitudinal component, necessary to be a good critical thinker (honest, confident, judicious, flexible and prudent), which when combined with the previous skills, support the psychologistic perspective of thinking ([10]).

The second perspective, the philosophical one ([1]; [2]; [17]; [31]), recognizes that critical thinking is sensitive to each field of knowledge and cultural context. The proposal is based on the development of cognitive resources supported by primary habits, consisting of assessment and evaluation of reasoning, development of argumentative processes, conceptual tools (basic knowledge of the field of knowledge and use and analysis of analogies and metaphors), and transfer of theoretical knowledge to practice; and secondary habits, which refer to having an open mind, curiosity, persistence and intellectual autonomy, and respect for opinions and group debate, aspects that coincide with the provisions raised by the psychological perspective.

Now, analyzing these two perspectives, it is clear that there are theoretical encounters between the philosophical and psychological perspectives, starting from the recognition that both positions see critical thinking as an essential process for decision making and problem solving; they also agree in recognizing the importance of evaluating evidence and arguments in a rational way and consider that critical thinking involves questioning assumptions and avoiding cognitive biases and focuses on the mental and cognitive processes that underlie critical thinking. In the same way, they propose that critical thinking allows us to identify how people develop and apply this thinking at different stages of life, taking into account the influence of factors such as memory, cognitive biases and metacognition in critical thinking ([19]).

In addressing the discussions between psychology and philosophy regarding critical thinking, a consensus was reached, in which, according to [12] ([12]), it is necessary to recognize and determine a series of skills and sub-skills within which constituent elements of the two perspectives are included. This consensus was published and completed in 1990, and was sponsored by the North American Philosophical Association with the Delphi project, lasting approximately two years. The participation of representatives from both areas sought to generate agreements regarding intellectual abilities and personal dispositions. This process was coordinated by Peter Facione, who, based on the writings, reflections and theories collected from each of the participants in the process, published an official document with the common elements generated during the process (see Table 1). It should be noted that this consensus (the Delphi project) is based on the six mental skills or mental processes suggested by psychology, and in addition to this, the development of argumentation based on the development of reasoning by philosophy is taken into account, which is necessary within the training of the critical thinker; [32] ([32]) in other words, aspects of the generic and domain-specific perspectives are unified.

This analysis allows us to infer that there are several aspects required to understand the theoretical and methodological bases of critical thinking, a fact that can be seen in different investigations which support the second aspect, in which it is identified that teachers at different levels of education have difficulties related to its comprehension and meaning ([23]; [32]; [37]; [34]). We suggest, within this research, to propose contributions aimed at the training of this group of academics based on the design of different types of educational interventions that promote theoretical and methodological knowledge, according to the disciplinary and cultural characteristics of the subjects and populations of the educational context.

Based on these discussions and analyses, science didactics ([34]) designed a proposal anchored to a group of categories, and taking into account the interests pursued here, the focus was directed towards problem solving, argumentation and metacognition (see Figure 1), recognizing the dependence both on the specific domain of knowledge as well as on the type of conscious and regulated knowledge raised from the psychologistic perspective, associated more with the development of metacognitive skills (self-regulation) than cognitive skills (inference, analysis, inference, evaluation and explanation); that is, it designs a synthetic model that integrates aspects of both theories (psychologistic and philosophical) that allows teachers to understand and transform the teaching and learning processes in the science classroom.

These observations generated interest in science education, since within the Ministry of National Education (MEN), it was proposed in recent decades that education contributes to the formation of critical thinking. However, different questions arose: On what theoretical and methodological perspective is it based? Do teachers have in-depth knowledge about critical thinking? These questions led to different investigations related to teacher training in science education, in which, once the analyses that answered the first question were carried out and they began to investigate and answer the second, a perspective was proposed in science education supported by three constituent elements: metacognition, problem solving and argumentation. It was found that for each of the constituents, there are also different theoretical and methodological developments, some of which are presented below:**Problem solving**: This is one of the cognitive activities of critical thinking, in addition to logical thinking, analysis, evaluation and responsible decision making with socio-scientific ([7]) or socially alive ([30]; [16]) problems such as climate change, biotechnology, cloning, etc. Some authors express that in problem solving, all the skills ([13]) and the cognitive resources of the subjects are enhanced (reasoning, arguments, analogies, metaphors, etc.).**Argumentation**: This is a social, dialogical and dialectical activity which contributes to the construction of school knowledge. In addition, it allows subjects to defend their ideas in discussions about different problems (socio-scientific or socially alive) and to choose between different models and theories to explain the phenomena of reality, supported by the use of evidence that supports the veracity of their postulates. In relation to the development of argumentative processes proposed by philosophy, the emphasis on the formal logical and pragmadialectical approaches is evident, where the coherence and deduction between the premises used by the subjects and the possible agreements that can be generated in a debate are proposed ([40]). Argumentation is, without a doubt, inherent to each field of knowledge, since it requires basic concepts, use of tests and evidence to support and sustain the premises, and ideas and explanations used in different contexts, in which science teaching has recognized contributions to the development of in-depth learning ([40]).**Metacognition**: This is a category that allows subjects to monitor and regulate their own cognitive processes. This category is closely related to self-regulation and enhances critical thinking in a central way in educational processes. “When speaking of a type of conditional knowledge, in addition to being aware of what must be done at a cognitive level, specific knowledge related to the task to be solved must also be used. In this sense, conditional knowledge brings together both cognitive and conceptual knowledge, and it is due to the importance of this interaction between the cognitive and the conceptual that conditional knowledge is especially important for the formation of critical thinking in students.” ([34]).

These analyses allow, in the first instance, the identification that although the self-regulation category, defined by the psychological line of research, and metacognition, addressed from the proposal in science didactics, agree in that they are important skills for the strengthening of thinking, for the former, self-regulation is of a generic nature, that is, it is addressed in any informal or formal context, while for the latter, metacognition is determined in the knowledge that is taught and learned in the classroom ([18]), which involves aspects related to the type of knowledge (declarative, procedural or strategic), regulation (planning, monitoring and evaluation of learning processes) and metacognitive awareness (purpose of the task and personal progress).

Another meeting takes place between the didactics of sciences and the philosophical perspective within the category of argumentation, understood as a primary habit. For these, argumentation allows the evaluation of the structure of the propositions and the coherence of the conclusion, that is, a logical-formal approach. On the contrary, in science didactics, this category is assumed to be a constitutive element of critical thinking from where the use of reasoning is enhanced; that is, it acts as an epistemic artifact that contributes to the construction of school knowledge.

Therefore, it is evident that in the three theoretical perspectives (psychology, philosophy and science didactics) the common point is the category of problem solving, differentiating that in psychology, it is a skill, in philosophy, it is a primary habit or stage of critical thinking and in science didactics, it is a constituent category; that is, it could be seen as the category that cross-cuts the three perspectives.

Once this analysis is carried out, it is inferred that there are several aspects required to understand the theoretical and methodological bases of critical thinking given its complexity, a fact that can be seen in different investigations in which it is identified that teachers at different levels of education have difficulties related to its comprehension and meaning ([23]; [32]; [37]; [34]). We suggest, within this research, to propose contributions aimed at the training of this group of academics, based on the design of different types of educational interventions that promote theoretical and methodological knowledge, according to the disciplinary and cultural characteristics of the subjects and populations of the educational context.

These reasons support this work, which could contribute to the discussion on critical thinking in teacher training in different areas, as well as for educational institutions that recognize a great need to reevaluate their classroom practices, giving the educational processes and interactions that today’s society demands and that suggest that training in critical thinking in classrooms is necessary the importance and relevance they deserve ([29]; [34]). For these purposes, the study was conducted with secondary school teachers in areas such as Ethics, Mathematics, Physical Education, Social Sciences and Spanish Language from an educational institution in the city of Manizales. In addition, it is possible that these aspects can be used in other studies as a guiding entity for research on teacher training and the development of critical thinking, recognizing the influence of the context as an aspect that permeates the teaching and learning processes in classrooms.

An example of this is evident in the studies on the training of university professors on the Colombian Caribbean Coast, belonging to semesters IV, V and VI, which mention that there is little understanding in the categories that consolidate critical thinking, which include arguing, inferring, proposing and solving problems. To address this situation, they propose an interactive program with some technological tools to contribute to the training of future teachers. Likewise, authors such as [36] ([36]) recognize that university students in countries such as Chile and Spain have difficulties in understanding critical thinking in depth, and to do so, they propose programs for the development of mental skills, recognizing that it is costly, in cognitive terms, to deploy all its constituent elements ([29]).

Other authors, such as ([26]), investigate the relationship and integration between the dispositions or predispositions that allow an attitudinal character to think critically, and the cognitive skills themselves, which involve the processes of critical reading in students and university teachers.

Studies on primary basic education teachers ([34]) identify two central aspects of the conceptions of critical thinking. The first recognizes that inquiry is the central category of the conceptions and teaching of critical thinking in the classroom; the second recognizes that, for the development of critical thinking, it is necessary to know the social context in which students operate, the development of arguments and the ability to reflect on society, relevant aspects in different research in science didactics. [37] ([37]) proposes that teachers must be clear about the concepts and methodologies for teaching critical thinking, and to do so, it is necessary to propose modifications to the study plans ([20]), since those that are designed and offered hinder their development; in other words, there is fear that what is learned in the classrooms may affect the development of the good critical thinker.

In secondary education, there are different works in countries such as Peru and Chile ([22]; [21]) on conceptions and knowledge about critical thinking, identifying not only deficiencies in the use and development of thinking, scientific inquiry and reasoning skills, but also a great need to design institutional spaces that allow the development of critical thinking strategies, dialogue and reflection skills.

However, research work at the local level in teacher training in basic secondary and middle education is incipient, given that the proposals put forward by the MEN (National Ministry of Education) in the study plans suggest emphasizing the development of mental abilities (psychological perspective) within each field of knowledge (philosophical perspective) in educational institutions, without making conceptual clarifications and precisions that allow understanding the potential and differences in each of the proposals which lead to identifying theoretical and methodological gaps in basic primary, secondary and middle levels. In addition to this, the proposal put forward by the didactics of sciences[note 1] is developed within some of the postgraduate and diploma courses offered by local universities, which suggests that the training of teachers is limited since it requires inter-institutional agreements with the secretaries of education to reduce the gap referred to in the literacy of teachers in critical thinking.

These analyses and other aspects can be seen in Table 2, where it is evident that authors from each perspective share the same ideas.

These are the reasons that lead us to want to answer the following question and the achievement of the following objectives:

How do secondary and high school teachers in the areas of Ethics, Mathematics, Physical Education, Social Sciences and Spanish at an educational institution in the city of Manizales, Colombia conceive and implement critical thinking in the classroom?

To understand how teachers in the areas of Ethics, Mathematics, Physical Education, Social Sciences and Spanish at an IE in the city of Manizales conceive and apply critical thinking in the classroom.To characterize the critical thinking of teachers in the areas of Ethics, Mathematics, Physical Education, Social Sciences and Spanish of an IE in the city of Manizales.

## 2. Methodology

This research is based on the qualitative approach, is descriptive–comprehensive, and uses as a method that of the case study ([6]; [8]; [33]), which incorporates different types of instruments and various data collection processes, such as participant observation, document analysis, surveys, questionnaires and/or interviews ([38]).

Five cases were selected from a population of 16 teachers (13 men and 3 women) whose ages ranged between 37 and 54 years and belonged to the areas of Ethics, Mathematics, Physical Education, Social Sciences and Spanish, of which one of them had completed doctoral studies (teacher 5), two master’s degrees (teachers 1 and 3), and the rest specialization studies (teachers 2 and 4) from an educational institution in the city of Manizales; none of the teachers had prior training in critical thinking issues. The criteria were first, teachers from different fields of knowledge; second, voluntary participation in the research proposal—in this case, five male teachers expressed their interest in the study; and third, the explicit recognition of the teachers in knowing the contributions of critical thinking from different perspectives on critical thinking in education and not having participated in any research on the phenomenon of critical thinking, thus contributing to better educational practices that allowed them to qualify their profession. These aspects were of interest for this study, given the different debates related to the generic nature and specific domain of critical thinking.

The information was collected through three instruments: a survey, an interview and a field diary. The survey consisted of seven questions which reflect the conceptions that each teacher contributes about critical thinking, the characteristics of a critical thinker, the specific and/or generic nature of thinking, the difficulties that arise in the classroom, the strategies used in the classroom, the skills that are identified in critical thinkers and the basic knowledge necessary to think critically in the specific area (see Table 3).

The second instrument, the semi-structured interview, delved deeper into the responses obtained in the previous survey, which allowed us to delve in a rigorous manner into what the teachers thought about the categories; that is, we delved deeper into the seven elements expressed by each teacher from their specific field of expertise (example teacher 4, Table 4).

The third instrument used was the field diary, in which the entire process carried out through classroom observation was recorded, mediated by audio and video recording. Once the information was collected with each of the three instruments, the data were analyzed, contrasting them with the different theoretical perspectives. For this, content analysis was used, which focused on the precisions of [4] ([4]), for whom content analysis is a set of methodological instruments applied to what he calls extremely diversified discourses (contents and containers).

### 2.1. Procedure

This research was carried out in three steps:

**1. Planning**: In this phase, teachers are approached and what the research process consists of is explained, the instruments to be used in the development of the event are mentioned, and the agenda to be followed is agreed upon according to the conditions and particularities of each teacher and the institution.

**2. Application**: In this study, the execution and application of the instruments is continued, with the structured survey being the initial basis for obtaining information on the phenomenon and with each teacher approached in a calm environment and with time available. Subsequently, an initial review and analysis of the information is carried out, with which the interview is designed, which is on the path to deepening the categories.

**3. Analysis:** At this stage of the research, the information is complete, and therefore the systematization, analysis, structuring and integration of the (5) hermeneutic units was carried out. The cycle is closed, the structured data are extracted through semantic networks and the information is triangulated and contrasted with the theory.

### 2.2. Analysis and Discussion

The analysis was carried out in relation to the interpretation and understanding of the six constituent elements by the five teachers: concepts, skills, characteristics, strategies, difficulties and their work in the classroom, information captured in semantic networks and their relationship with the different authors who study critical thinking. Given the volume of information, the semantic networks of the five teachers are presented below and a collective analysis will be carried out later.

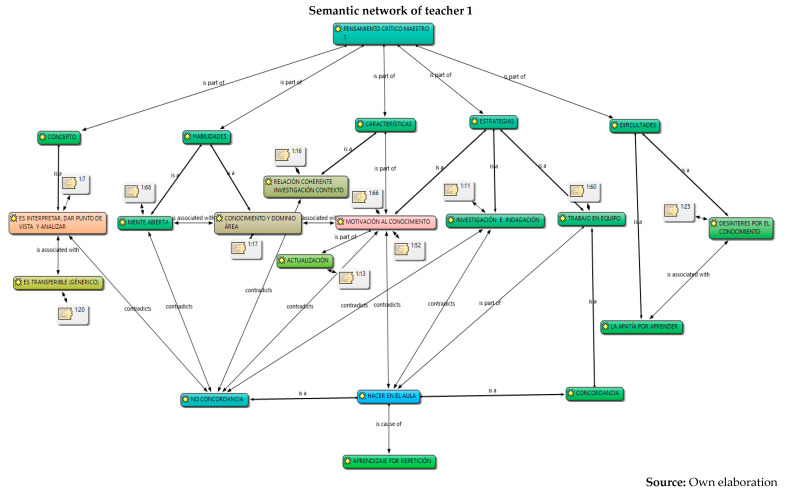


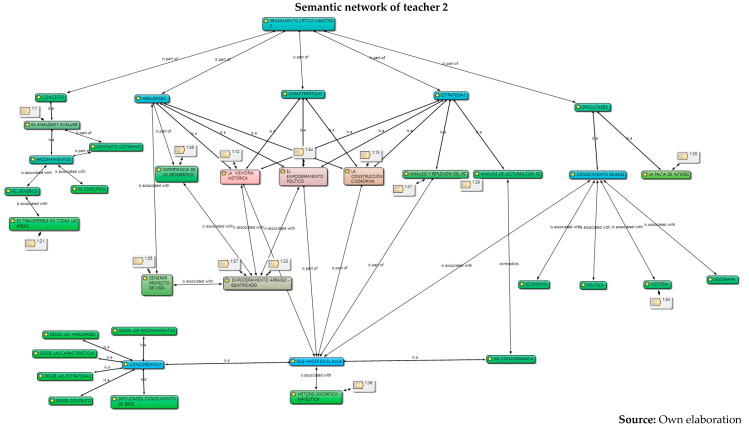


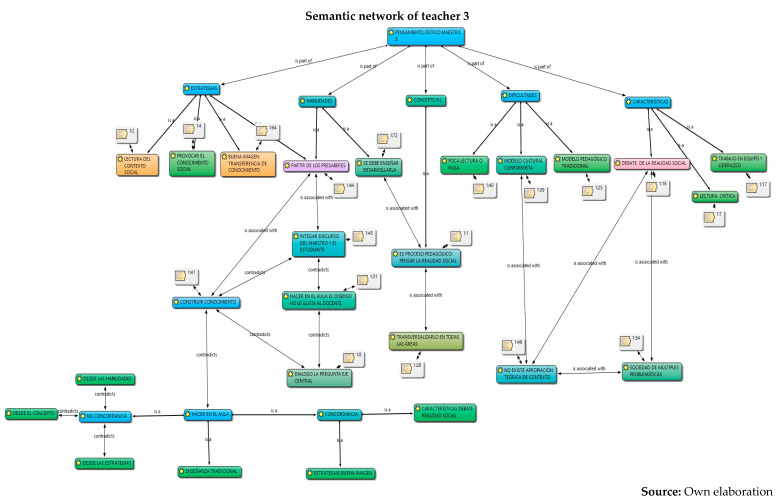


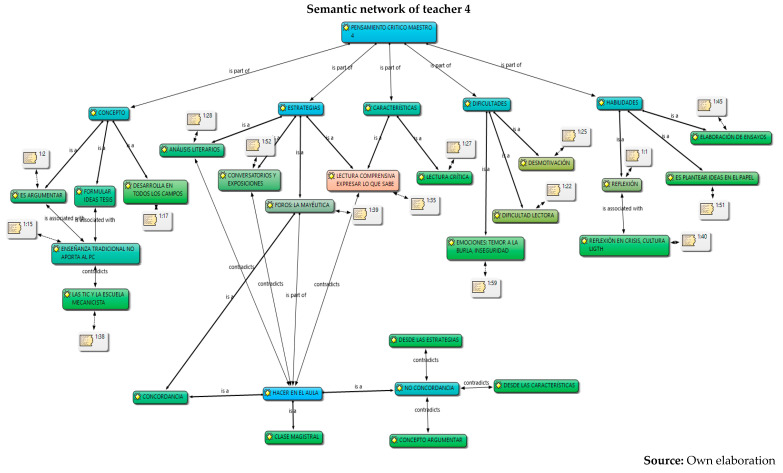


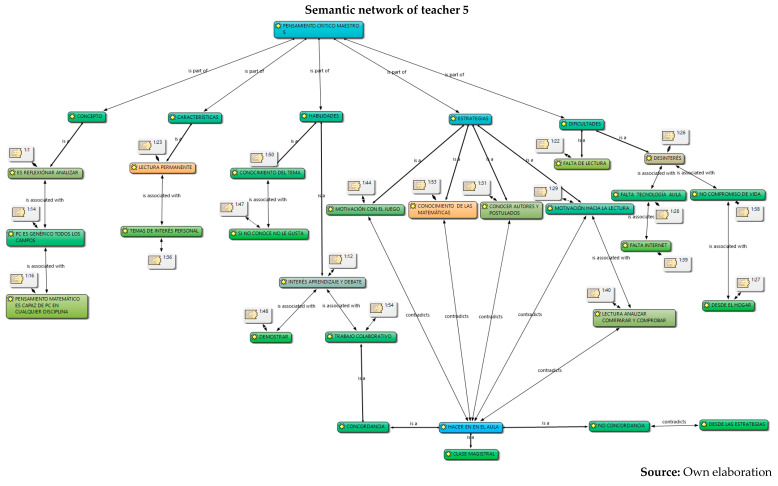


Below is an analysis of the five teachers in each of the categories (conceptualization, skills, characteristics, strategies, and difficulties and work in the classroom), according to the content referred to in relation to critical thinking and its constituent elements. At first, it can be concluded that there are several conceptions regarding the subject; that is, there is no unification on the part of the collective regarding critical thinking. However, the following were identified:From the conceptualizations, the analysis category and from different approaches: critical thinking, contextual and reasoned, is the idea that presents greater relevance in the five teachers, as it is part of both conceptualization and skills. For example, their responses were as follows:

M1: “I consider that critical thinking is a structured and critical look that presupposes analysis.”

M2: “The fundamental reason is because we must begin to analyze everyday contexts, everyday life, begin to analyze everything that has to do with social reality.”

M5: “In mathematics, everything we see has already been written, analyzed, invented, reasoned, criticized, everything, everything has been done, yes, what do we do? What we do is simply take that up again and analyze it.”

The ideas of teachers 1, 2 and 5 are related to the approaches presented from the psychological perspective ([5]; [10]; [9]), which leads to identifying a trend of this category in the thinking of teachers. It is possible that this can be part of their actions in the teaching and learning processes of their students; in addition, this skill is also mentioned in the studies carried out on primary school teachers ([34]) and in the approaches of [12] ([12]). It is part of the consensus between the fields of psychology and philosophy and it should be noted that the analysis of the context and reasoning is studied from the philosophical perspective ([17]; [2]), which allows us to infer that teachers mix elements of both perspectives, possibly due to a lack of knowledge about critical thinking.

2.Interpretation and reflection are two other categories that are part of the concepts of critical thinking, as evidenced in the following responses:

M1: “Seeking to organize knowledge and interpret it and investigate how to understand it and if possible contribute to its improvement and construction.”

M5: “Comment on things, talk about things and then start to reflect on whether what has been said is true, or not true, or if it can be improved, what we can do with it”

Their answers are part of the proposals of some authors such as [10] ([10]) and [9] ([9]), who suggest that understanding meaning is the first skill (interpretation) necessary for the sequential development of the others. The second skill, reflection, is related to the evaluation and assessment of ideas (reasoning and arguments), a category studied from a philosophical perspective ([17]; [3]), and is, in turn, one of the agreements expressed in the work of [12] ([12]) in the Delphi project. However, when they were asked specifically about the skills of a critical thinker, the teachers presented different types of answers, in which they also mention dispositions; that is, the teachers consider both constituent elements in a single category, where conceptual precisions are required for their understanding.

3.Faced with the characteristics necessary to think critically, secondary school teachers place three postulates on the subject; that is, from the philosophical perspective, the answers of M1, M2, M3 are located as follows:

M1: “Be clear about the importance of research and the influence of the area within the particular and everyday context.”

M2: “We can have people who behave in an acceptable manner in society, who comply with the rules and who defend their rights…we will have people with civic culture”

M3: “Engage in a process of dialogue with the student and not impose knowledge that is suddenly useless, then education must start from that didactic, methodological and pedagogical principle”

The ideas of teacher 1 approach some elements that support the philosophical perspective, and that, according to ([17]): “The critical thinker is skilled and responsible, uses good judgment, since it is based on criteria, corrects themselves, and is sensitive to the context” (p. 146).

It can be said that teacher 2 is close to what is stated in the psychological perspective, supported by authors such as [10] ([10]) and [9] ([9]), for whom critical thinking contributes to the formation of good citizens. At the same time, it makes sense with the didactics of sciences, because according to [34] ([34]), “one of the challenges of pedagogy is the reconstruction, the redefinition of the experiences that humanity has gone through and that are directly linked to the intentionality of the formation of citizens” (p. 93).

In relation to the approaches of teacher 3, dialogue with students is one of the elements that are part of the approaches of the dialectical approach to argumentation ([2]), because it contributes to the construction of reasoning and allows exploring and understanding the students’ explanatory models on different study phenomena, in addition to enhancing communicative processes that contribute to the co-construction of knowledge in classrooms ([40]).

4.Within the strategies, motivation is considered by teachers to be the central category, because it is also part of the skills that a critical thinker must possess. Some of their answers are as follows:

M1: “The motivation towards research and its importance, not only in our area, but in all areas of knowledge.”

M5: “ Try to permanently motivate students to read and motivate them through games”

This category is also of great relevance for primary school teachers, however, it is focused on the students and not on the teacher’s actions ([34]); in other words, students must be motivated intrinsically by themselves and not by the design of spaces that allow motivation ([14]), and act on how, why and for what to control their emotions in their learning ([28]).

5.Regarding difficulties, the five teachers agreed that the category of disinterest is the one that is most prevalent in classrooms and that it directly impedes the formation and development of critical thinking. In this regard, they stated the following:

M1: “…if there is no interest in investigating, in acquiring new knowledge, I believe that critical thinking will not exist.”

M2: “What happens is that young people are showing serious problems of interest, of attitude towards life, one notices, the same indifference is terrifying and one wonders what is happening to them.”

This is undoubtedly a highly relevant category within science teaching, and authors such as [28] ([28]) recognize that the emotional states generated by teachers in classrooms affect students’ motivation and interest, and can interfere with academic performance levels, commitment and learning actions in the classroom. Likewise, interest can be one of the triggers of curiosity, which is linked to the desire for new information and learning goals; in other words, it contributes to intrinsic motivation for the taste and pleasure of learning the subjects of study.

6.Once the analysis of their conceptions on critical thinking has been carried out, in which conceptual dispersion is observed in the different components of the research (conceptualization, skills, difficulties and characteristics) in the five teachers, the coherence with the strategies proposed for their development in the classroom is analyzed below, and for this, the observation of their work in the classroom was carried out through the field diary, evidencing the following.

The category of motivation is considered to be the trigger category among the strategies proposed by the secondary school teachers of the La Sultana Educational Institution of Manizales, and this is also part of the skills that a critical thinker must possess. This category is assumed to be an essential part in the development and training of the critical thinker, and despite being very relevant in the strategies, it is assumed as a motivational element that falls on the shoulders of the students and not on the actions of the teacher ([34]). It is possible that this is due to lack of knowledge on the subject, since it is also mentioned within the difficulties that arise in the classroom; in this regard, it is found that this strategy is shared by M1, M3 and M5.

In some of the research in science didactics, when using the terms of [34] ([34]), for primary school teachers in Manizales, motivation in the classroom takes on total relevance and meaning, and this must be fostered by the teacher. At the same time, [5] ([5]) maintained that good stimuli must be generated in classroom interactions mediated by teaching processes, which contribute to a better educational act in the classroom.

However, the use of the traditional model by M1, M3, M4 and M5 in their classroom practices was observed, in which the transmission of knowledge is the protagonist. That is to say, this occurs although these teachers outline motivational elements that are articulated with important discursive elements in the configuration, conceptualization and enhancement of critical thinking, from the elements that constitute it (skills, characteristics, strategies and difficulties) and with respect to substantial categories that are developed in the classroom such as analysis, reflection, argumentation, open mind, debate, dialogue, maieutics and reading of the context, all elements that can be located in the three perspectives: psychological, philosophical, science didactics.

At the same time, there is coherence in teacher 2 within what he proposes in his speech and in his practice; that is, he is a teacher for whom it is important to contextualize the students in the subject to be discussed, as the teacher presents his speech. He constantly asks questions to his students and involves them in class; therefore, he makes use of maieutics throughout the development of his classes (Socratic method); also, he carries out activities with the use of technological tools in the same way. The activities proposed by teacher 2 are aimed at young people generating their own positions and according to their preferences; that is, they promote the construction of reasoning together with their students. In this regard, this resembles [25] ([25]), who expresses that Socrates had more questions than answers and many times the dialogue ended in perplexity, very far from an absolute definition, and that he also preferred to make his friends “companions” in the long and uncertain path of definitions, having “self-knowledge”.

In other words, his class is totally consistent with his speech and the need to forge in his students their own opinion criteria, with the research and inquiry of the proposed topics. In general, this presents an agreement regarding the development of critical thinking from the components that configure it (concepts, skills, characteristics, strategies, difficulties and what to do in the classroom). Finally, as a conclusion, it can be inferred that teacher 2, despite conceiving critical thinking in two directions, psychological and philosophical, has postulates which focus on the line of research from philosophy, [3] ([3]), where the mastery of the disciplinary field is the foundation for the development of critical thinking. It can also be considered that his doctoral level in political science allows him to emphasize the topics of his area in a way that supports the development of critical thinking from its specificity ([34]), and his didactic orientation is inclined to the development of questions to motivate the student.

## 3. Conclusions


Different conceptions of the five secondary school teachers about critical thinking were identified and understood. These refer, firstly, to the concept or conceptualization; secondly, to different constituent elements that refer to the skills, their characteristics; and thirdly, to the strategies and difficulties that teachers find to develop in their classrooms. From these aspects, it was possible to interpret that the critical thinking of teachers presents, firstly, conceptual dispersion in their responses; that is, teachers mix different elements of theoretical perspectives within their conceptions. This may be due to the different proposals that exist from the fields of knowledge of psychology and philosophy. However, it is evident that there is a general lack of knowledge about the proposal in science didactics, since they barely manage to mention some aspects of the three dimensions or constituent elements, despite having around 10 years of experience in their publications.Furthermore, it is evident that the relationship between their conceptions and their classroom practices, in general, is incoherent. One of the five teachers puts his declarative knowledge into practice; it is possible that his academic training (doctoral) influences his theoretical and methodological training, an aspect that is not reflected in the other four teachers, where the incoherence and traditional teaching models are clear. Based on these analyses, it is recommended to carry out training processes in each of the theoretical perspectives of critical thinking, which would allow specific critical thinking in teachers, since many ideas are interwoven that could generate epistemological and didactic obstacles when teaching study topics.The critical thinking of the five teachers was characterized based on different constituent elements: conceptualization, skills, characteristics, strategies and difficulties. In their alternative conceptions, these could be structured into personal models or explanatory models ([40]; [11]; [15]). From this perspective, it is understood as an amalgamated explanatory model, since it integrates different aspects of the psychological and philosophical perspectives on critical thinking. This could be identified in subsequent research on the critical thinking of teachers in primary, secondary and middle basic education. To this end, it is necessary to implement programs that integrate the three perspectives (psychological, philosophical and science didactics) as a possible contribution to breaking traditional educational perspectives, which, as was evident, continue to prevail in four of the five teachers who participated in this research.Furthermore, it is necessary to strengthen the relationship between critical thinking and internal institutional dynamics in order to strengthen classroom processes and go through them with consequent and self-regulated cognitive processes, which contribute to the construction of knowledge, and which provide effectiveness in the didactic and pedagogical components permeated by the curricula, visions and institutional missions. This allows the articulation of the discursive elements of teachers with the pedagogical and didactic processes of the classroom in order for them to form and develop critical thinking with their students, through the configuration of conscious and self-regulated spaces that provide support, planning, execution, monitoring and evaluation of the processes in order to promote critical thinking in educational institutions.


## Figures and Tables

**Figure 1 jintelligence-13-00037-f001:**
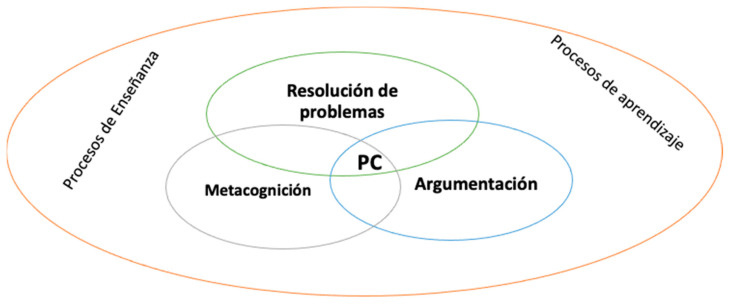
Constituent elements of critical thinking in science education. Source: taken from [34] ([34]).

**Table 1 jintelligence-13-00037-t001:** Critical thinking skills and sub-skills.

Skill	Sub Skills
Interpretation	Categorization, decoding and clarification of meanings
Analysis	Examine ideas, identify arguments and analyze arguments
Inference	Examine evidence, conjecture alternatives and draw conclusions
Assessment	Evaluate statements and arguments
Explanation	Knowing how to argue and raise agreements or disagreements
Self-regulation	State results, justify procedures, present arguments, self-examine and self-correct

Source: Taken from [12] ([12]).

**Table 2 jintelligence-13-00037-t002:** Fields of knowledge, components, difficulties and similarities of critical thinking in psychology, philosophy and didactic sciences.

Critical Thinking	Field of Knowledge		
Components	Psychology	Philosophy	Science Didactics
Concept	Regulated transference thinking	Thinking that develops judgments sensitive to each field of knowledge	Integrated synthetic model: problem solving, argumentation and metacognition.
Focus	Development of mental skills and dispositions	Development of criteria (primary and secondary habits)	Integration of constituent elements: problem solving, argumentation and metacognition.
Skills, criteria (primary habits) and constituent elements	Skills: interpretation, analysis, inference, evaluation, explanation and self-regulation. Dispositions	Primary habits: arguments, relationship between theory and practice and conceptual tools (background knowledge, metaphors and analogies)	Integrated elements: problem solving, argumentation and metacognition. Constituent elements in the design of teaching units
Characteristics or dispositions, secondary habits and constituent elements	Trust in reason, facing one’s own predispositions, propensity to make judgments, mental flexibility and intellectual courage	Open mind, intellectual curiosity, intellectual autonomy, intellectual persistence and respect for the discussion group	Emotional motivational component in science learning: surprise, confusion, curiosity, interest-type enjoyment and anxiety.
Strategies	Analysis and self-regulation in different activities, and critical reading and problem solving through the development of mental skills.	Reading the social and disciplinary context and using logic in the construction of reasoning.	Diseño y resolución de problemas reales, auténticos, sociocientíficos a partir del uso de actividades epistémicas: dialogo razonado, debate
Difficulties	Lack of training in both people and curricula, teachers in the development of both skills, and attitudes on critical thinking	For the construction of reasoning, basic knowledge of each field of knowledge is necessary. Obstacles to knowledge: opinion, basic experience, ease, simple rationality, etc.	Educational obstacles in teaching and learning processes. There is a lack of knowledge about critical thinking in teacher training.
**Meeting Points**			
1. Problem solving	All mental skills are enhanced in problem solving	All cognitive resources (primary and secondary habits) are used in problem solving	It is a constituent element and stage of critical thinking.
2. Attitudinal component	Called dispositions and/or characteristics	They refer to secondary habits	It is oriented under the emotional–motivational approach that affects the teaching and learning processes.
3. Consensus in the Delphi project	A type of reflective, self-regulated thinking that uses skills and sub-skills, including analysis and evaluation of arguments.	It is a thinking that uses the development of reasoning and critical judgment, which uses skills and sub-skills, including analysis and evaluation of arguments.	Type of reflective, reasoned and regulated thinking necessary in teaching and learning processes. It recognizes argumentation and metacognition within its categorical integration.

**Table 3 jintelligence-13-00037-t003:** Types of questions.

Types of Questions	
1.	What do you mean by critical thinking?
2.	Mention and justify 3 characteristics that are necessary for critical thinking in your disciplinary area?
3.	Do you consider that critical thinking is sensitive to the field of particular specific domains, or, on the contrary, can it be transferred to all fields, (those who think critically in mathematics do the same in language, art or social sciences, among others)? Justify your answer.
4.	Mention and explain the 3 main difficulties you have in developing critical thinking in your classroom.
5.	List and explain the top 5 strategies you use to develop critical thinking in your classroom.
6.	List and explain what skills are necessary to think critically in your discipline.
7.	Do you think that basic knowledge is necessary to think critically in your field? Why? Mention and explain the basic knowledge necessary to think critically in your disciplinary field.

Source: author’s elaboration.

**Table 4 jintelligence-13-00037-t004:** Types of interview questions posed to Teacher 4.

Types of Questions	
1.	Why do you think reading short texts develops critical thinking?
2.	Why do you think literary analysis is important for developing critical thinking in your classroom?
3.	Why do reading comprehension workshops help strengthen critical thinking?
4.	Why do you think forums and debates are important for developing critical thinking in your field?
5.	How do discussions and presentations contribute to critical thinking in your field?

Source: author’s elaboration.

## Data Availability

The raw data supporting the conclusions of this article will be made available by the authors on request.

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
