# Peer review of "Critical Thinking and Teacher Training in Secondary Education"

_jintelligence, 2025, doi:10.3390/jintelligence13030037_

Round 1
Reviewer 1 Report
Comments and Suggestions for Authors
Critical Thinking and Teacher Training in Secondary Education
The objective of this study is to characterize how secondary and high school teachers in the areas of Ethics, Mathematics, Physical Education, Social Sciences, and Spanish in an educational institution in the city of Manizales, Colombia, conceptualize and implement critical thinking in the classroom.
The study is both relevant and timely. As the authors point out, the development of critical thinking is absolutely essential in education—not only for its contribution to cognitive development but also because of its psychosocial importance. In fact, it has become one of the central and cross-disciplinary objectives in teacher training at all educational levels. However, as the authors also note, there is insufficient understanding and training among teachers, both from a conceptual perspective and in terms of its implementation in classrooms.
The introduction of the study is clear, adequate, and current, and the objectives are well-defined.
Method
While the study faces the limitations inherent in case studies, it also presents some of their advantages by providing valuable information that could inform more methodologically robust future research. Even so, there are aspects where the authors could improve the text:
Areas for Improvement
Introduction
Given that the study is highly contextualized within a specific country, educational system, and location, the authors should perhaps be more explicit in discussing how critical thinking development is addressed in the secondary and high school curricula in their country, and how it is considered within university study programs—at least in the programs completed by the participants. Furthermore, it is unclear whether postgraduate or continuing education programs exist in this context that might address the training gaps of in-service secondary and high school teachers.
Participants
Some aspects of the study’s participants remain unclear. For instance, have they had comparable training opportunities about critical thinking? Otherwise, this variable could make the sample more heterogeneous in terms of their foundational knowledge of critical thinking, in addition to the existing didactic and scientific differences across their respective subjects.
It would also be beneficial to expand the description of how the five specific teachers were selected, including details about their gender, age, academic qualifications (undergraduate and postgraduate), whether they taught at the secondary or high school level, and the criteria used to determine their representativeness of the target population (16 teachers). The gender of the participating teachers is also not described, which could be relevant to analyze whether it influences the studied aspects (conceptions, classroom implementation, etc.).
Formal Aspects
From a formal standpoint, the authors do not correctly follow APA 7th edition guidelines for in-text citations. This issue is particularly evident in citations with more than two authors and in their alphabetical ordering within parentheses. Additionally, it would be preferable for the article’s structure to follow standard sections such as: Method (not Methodology), Participants, Instruments, Procedure, Analysis, and Discussion, instead of the current format: Methodology (Research Phases and Analysis and Results) and Conclusions.
References
Finally, there are some errors in the references that should be addressed:
- References 5, 15, 25, and 28: Italics are missing for the journal name.
- Reference 8: Missing author and year.
- References 18 and 32: The journal issue number should not be in parentheses.
- Reference 20: The volume number should be italicized.
- Reference 36: The issue number should not be italicized.
- Reference 22: The initials of the authors' names are missing periods; check for similar cases throughout.
- References sometimes use & and other times and; this should be standardized.
- Reference 27: This appears in English; it is not the spanish versión?, remove the city name (unnecessary). The Publisher is "Pirámide").
- Reference 29: Incorrect use of italics; the reference must be properly formatted.
In conclusion, despite the generalization limitations of a case study, the qualitative results presented provide useful insights. With the recommended revisions, the study is suitable for publication.
Author Response
Reviewer 1
It is argued that while the method is fine, even so, there are aspects where the authors could improve the text; In light of this, the method and criteria for selecting the sample were further expanded.
The references referred to were organized according to APA standards and the information was completed.
In relation to the recommendation on the introduction, the following text was incorporated to respond to the request: "Now, in the case of Colombia, critical thinking is adopted in formal education as a reference for quality, however, it lacks recognition in its epistemological, pedagogical and methodological foundations, by limiting the teaching practice in the classroom, given that it works from notions, beliefs and subjective perspectives of what it refers to. This issue led us to think about the need to work on the development of critical thinking with teachers, as a commitment to contribute to quality education from the classrooms."
Regarding the selection of participants, the information on the selection criteria was expanded and it was clarified that they did not have training in PC, the incorporated text was: "Five cases are selected from a population of 16 teachers, (13 men and 3 women) whose ages range between 37 and 54 years and belonging to the areas of: Ethics, Mathematics, Physical Education, Social Sciences and Spanish, of which one of them has doctoral studies (teacher five), two master's degrees (teachers one and three) and the others with specialization studies (teachers two and four) from an Educational Institution in the city of Manizales; none of the teachers had prior training in critical thinking issues. The first, teachers from different fields of knowledge; second, voluntary participation in the research proposal, in this case, five male teachers expressed their interest in the study; third, the explicit recognition of the teachers in knowing the contributions of critical thinking from different perspectives on critical thinking in education and not having "They have not participated in any research on the phenomenon of critical thinking, and thus, contribute to better educational practices that allow them to qualify their profession. These aspects were of interest to this study, given the different debates related to the generic nature and specific domain of critical thinking."
Finally, regarding formal aspects, the references were organized according to APA standards, as can be seen in the document.
Reviewer 2 Report
Comments and Suggestions for Authors
The theme of this study is interesting. There are however conceptual and methodological problems that I describe thoroughly in my comments in the attached manuscript.
Briefly, the distinctions among the three perspectives of critical thinking (as you call them) are not clear. This fact together with the lack of a table that shows how the three different perspectives of critical thinking are identified in relation to the six elements you are looking for in teachers' thinking and practice about critical thinking (concept, skills, characteristics, strategies, difficulties and strategies) cause problems in the reliability of the research.
The reader is not clear whether you are supporting a synthetic model of critical thinking skills and attitudes or are you pointing out the epistemological and practical consequences of adopting a different perspective on critical thinking. In any case, the epistemological and practical consequences of adopting a different perspective on critical thinking are not clear.
You refer to content analysis based on the three different perspectives but as I mentioned above we do not have a table of the categories that fit in each one of these three perspectives regarding concept, characteristics and all the other elements you are looking for to see how you ended up with your analysis and conclusions.
You later mention semantic network analysis but this is presented in Spanish and one can not understand if not a Spanish speaker. The results of this semantic network analysis are only shown in graphs but they are not actually presented in the results section. Rather you choose to report on the results for each element (e.g. concept or characteristics of critical thinking) rather than show how these elements relate to each other as shown in the network.
While you refer to observation of teacher practice, data are presented very shortly and only at the end of the findings. The analysis of observation data is not presented in an adequate way.
Findings should be better presented in my opinion, possible with the use of tables to better organize the information and after taking into account the comments in the manuscript.
Overall, the contribution of this study in the field of teachers' critical thinking and practice is not clear. Major changes are needed to clarify critical thinking conceptually and the ways data were analysed as well as what is the new information that this study brings in to the field of critical thinking.
I hope you will find these comments useful for the improvement of your manuscript.

English language should be improved because there are quite many cases in which the meaning is not clear. Long sentences also do not help in the clarity of meaning.
Author Response
Se han realizado los ajustes

Reviewer 3 Report
Comments and Suggestions for Authors
The article is original, interesting and particularly relevant to initial teacher education.
The abstract clearly presents the research problem, the aim, but it would be suggested to present research method applied, as well as main conclusions.
Introductory part is rich in citations, arguing about different approaches for critical thinking. However the differences between psychological and philosophical approaches are not sufficiently highlighted. Rather the opposite. For example, enviromental and cultural contexts are important for both. Philosophical approach views critical thinking from ontological perspective and raises existencial questions, rather than instrumental. It woud be reccomended to review this part and support it with more sound arguments.
Methodological part is quite clear. The case study looks relevant method to disclose the problem under investigation. The representation of project results in pictures must be done in English.
Conclusions are in line with the whole text. However, it is recommneded to present the limitations of the article, as well as reccomendation for the future studies.
Author Response
The evaluator states that, “However the differences between psychological and philosophical approaches are not sufficiently highlighted. Rather the opposite. For example, enviromental and cultural contexts are important for both. Philosophical approach views critical thinking from ontological perspective and raises existencial questions, rather than instrumental” In this regard, an extension was included on the similarities between both perspectives, and it is stated that, “The other thing is that more than identifying differences in the work, we were interested in starting from the meetings; For this reason, the following was incorporated into the text: "theorists between the philosophical and psychological perspectives, starting from the recognition that both positions see critical thinking as an essential process for decision-making and problem solving; they also agree in recognizing the importance of evaluating evidence and arguments rationally and consider that critical thinking involves questioning assumptions and avoiding cognitive biases and focuses on the mental and cognitive processes that underlie critical thinking. Likewise, they propose that critical thinking allows us to identify how people develop and apply this thinking at different stages of life, taking into account the influence of factors such as memory, cognitive biases, and metacognition on critical thinking."
The images are organized in English language.
Reviewer 4 Report
Comments and Suggestions for Authors
The topic is relevant and topical but I consider that the sample of 5 participants is not enough to reach relevant conclusions.
I suggest that you expand the sample to give more solidity to your research.
Author Response
The evaluator explains that, The topic is relevant and topical but I consider that the sample of 5 participants is not enough to reach relevant conclusions. Since the methodology is based on case studies, the topic of participant selection and the reasons that give weight to the work are expanded, which was carried out with a follow-up of one school year.
Since the methodology is based on case studies, the topic of the selection of participants and the reasons that give weight to the work are expanded, which was done with a follow-up of one school year. For this reason, the following was expanded in the methodology. Participants: Five cases are selected from a population of 16 teachers (13 men and 3 women) whose ages range between 37 and 54 years and belong to the areas of: Ethics, Mathematics, Physical Education, Social Sciences and Spanish, of which one of them has doctoral studies (teacher five), two master's degrees (teachers one and three) and the rest with specialization studies (teachers two and four) from an Educational Institution in the city of Manizales; none of the teachers had prior training in critical thinking issues. The first, teachers from different fields of knowledge; second, voluntary participation in the research proposal, in this case, five male teachers expressed their interest in the study; third, the explicit recognition of the teachers in knowing the contributions of critical thinking from different perspectives on critical thinking in education and not having participated in any research on the phenomenon of critical thinking, and thus, contribute to better educational practices that allow them to qualify their profession. These aspects were of interest for this study, given the different debates related to the generic nature and specific domain of critical thinking.
Round 2
Reviewer 1 Report
Comments and Suggestions for Authors
The authors have adequately addressed the recommendations provided in the initial review. In particular, they have responded to the suggestions regarding the theoretical introduction and the study's methodology. The contributions made in the discussion have further enriched the article, and the formal aspects related to citations and bibliographic references have also been corrected. The manuscript has been significantly improved in its current version, maintaining its relevance, timeliness, and scholarly interest. For these reasons, I consider that it is now suitable for publication in its present form.
However, during the editorial process, the authors should ensure consistency in the citation format of DOIs, as different styles are currently used in the references. Additionally, in lines 431 and 432, the term 'difficulties' appears twice within the same phrase ('conceptualization, skills, difficulties, characteristics, difficulties') and should be revised to improve clarity.
Reviewer 2 Report
Comments and Suggestions for Authors
The authors have responded adequately to the review and I agree for the paper to be published.
Reviewer 3 Report
Comments and Suggestions for Authors
The authors have improved the paper and it looks much better know,
Reviewer 4 Report
Comments and Suggestions for Authors
The recommended modifications make the research work more robust.